# Yield and Fruit Characteristics of Tomato Crops Grown with Mineral Macronutrients: Impact of Organo-Mineral Fertilizers through Foliar or Soil Applications

**DOI:** 10.3390/plants13111458

**Published:** 2024-05-24

**Authors:** Grazia Disciglio, Annalisa Tarantino, Laura Frabboni

**Affiliations:** Department of Agriculture, Food, Natural Resources and Engineering (DAFNE), University of Foggia, Via Napoli 25, 71122 Foggia, Italy; annalisa.tarantino@unifg.it (A.T.); laura.frabboni@unifg.it (L.F.)

**Keywords:** tomato, biostimulant, humic and fulvic acids, SPAD, yield

## Abstract

The utilization of plant biostimulants has gained importance as a strategy by which to improve plant productivity and soil health. Two independent trials were conducted across two seasons (2021 and 2023) to evaluate the effects of foliar or soil applications of various commercial organo-mineral fertilizers (Futuroot^®^, Radicon^®^ Amifort^®^) with biostimulant action that is exerted on yield and fruit characteristics of processing tomato crops (cv Taylor F1) that have been exposed to mineral macronutrients. These treatments were administered three times during the season: at the transplanting, pre-flowering and berry development stages. In the first trial, conducted in two fields characterized respectively by low and high fertility, foliar applications of Radicon^®^, which is based on humic acid and amino acids, increased the leaf greenness index SPAD compared with the control. Furthermore, the leaf green colour intensity (SPAD index), measured during the reproductive phases of the tomato exhibited a positive correlation (R^2^ = 0.726) with the marketable yield obtained. This increase in marketable yield was significant in the biostimulant treatment compared with the control in both soils, especially in the soil characterized by lower fertility (16.1%), when compared with the more fertile soil (6.8%). In the second trial, conducted in the low-fertility field mentioned above, soil applications of all biostimulants (Futuroot^®^, Radicon^®^ and the combinations [Radicon^®^ + Amifort-Plus^®^]) significantly increased the marketable yield by 27.8%, 13.5% and 27.7%, respectively, compared with the control. The most significant beneficial effects of both Futuroot^®^ and [Radicon^®^ + Amifort^®^] could be attributed to the combination of humic acids and auxins, cytokinins or microelements (Zn, Mn, MgO) present in the formulation of these products. Furthermore, the increase in marketable yield obtained when Radicon^®^ was applied to leaves was higher (16.1%) than that observed with soil application (13.5%). In both trials, no relevant effects of biostimulant products were observed on most of the physicochemical characteristics of tomato fruits. In conclusion, the biostimulants based on humic acid and amino acids combined with chemical fertilizers tested in the present study and applied by fertigation were more effective in improving tomato yield, and therefore they can be recommended for efficient agricultural production.

## 1. Introduction

Tomato (*Lycopersicon esculentum* Mill.) is the vegetable crop with the highest demand and greatest economic value worldwide. Therefore, it is one of the most cultivated horticultural crops globally [1]. In 2022, the annual world tomato yield reached 186.821 million metric tonnes over a cultivated area of 4,917,735 hectares [2]. In the province of Foggia, the cultivation of processing tomatoes holds the Italian record, covering an area of over 15,000 hectares and boasting a production of approximately 1.425 million tons. This represents 89% and 25% of Apulian and national production, respectively [3].

The current method of fertilizing tomato in mineral intensive forms of agriculture primarily relies on the application of chemical fertilizers, rather than organic. As a result, most soils are deficient in organic matter [4]. Therefore, the integration of inorganic chemical fertilizers with sources of organic fertilization, such as organo-mineral fertilizers with biostimulant action, becomes crucial. These agricultural biostimulants (Abs) are commercial products based on a wide range of compounds, including humic and fulvic acids, hydrolysed protein and nitrogen-containing compounds, seaweeds and plant extracts, chitosan and other biopolymers, and beneficial microorganisms [5,6]. Biostimulants are becoming increasingly significant in agriculture, being considered environmentally sustainable and economically favourable solutions to the optimization of crop productivity. In recent time they have become an increasingly popular tool as a source of plant nutrition, either as an alternative to synthetic fertilizers or in conjunction with them. Their effects are still largely unknown today, but they typically have a positive effect on plants [7,8,9]. Their action on plants is exerted through several mechanisms, including hormone-like activity production, enhancement of photosynthesis, and the promotion of plant–soil microorganism activity [10,11]. Furthermore, given the value consumers place on hedonic measures of fruits and vegetables, there is interest in strategies that farmers can use to improve these quality measures of their tomato fruits [12]. Particularly, amino acid and humic acid compounds, derived from the decomposition of plant and animal substances, play a significant role in optimizing the physical and biological properties of agricultural soils, managing nutrient availability and improving the growth and metabolism of some plants [13,14]. The current use of these substances, initially applied exclusively to the soil, now includes foliar application [15,16,17], with the aim of stimulating natural processes that enhance nutrient absorption and utilization efficiency [18]. Their action on soil nutrient availability and uptake has been attributed to several mechanisms affecting soil processes and plant physiology [15,19,20,21]. Furthermore, their effectiveness is highly influenced by soil fertility conditions and is pronounced under conditions of poor fertility and low organic matter content [5]. Numerous studies conducted on tomato crops in a controlled environment or pods [22,23,24,25,26,27,28,29,30,31,32] have reported positive influenced on yield and on morphological and physiological parameters. However, reports on the potential of biostimulants in the field are less explored. This is mainly due to the variety of underlying factors in crop fields, including weather variability, climate fluctuations, soil type, and field management [14]. Furthermore, the application of biostimulants has recently been extensively studied on tomato crops under abiotic stress conditions (temperature, drought, salinity, nutrition) to enhances growth, improve yield and quality, and minimize the negative effects of stress [23,33,34,35,36,37,38]. Several studies have also been conducted under mineral nutritional conditions [17,39,40], but comparative research on fields of different levels of fertility is lacking.

The aim of this research was to evaluate the potential use of different organo-mineral fertilizers, mainly based on humic acids and amino acids, applied via foliar or soil fertigation, on the yield and characteristics of the fruit of processing tomatoes grown in two soils with different fertility levels, under a standard level of synthetic fertilizer.

## 2. Materials and Methods

The research was structured as shown in Table 1.

### 2.1. Site Description and Experimental Setup

Two open-field separate trials were conducted in Foggia province (Apulia region, southern Italy) to assess the efficacy of various organo-mineral fertilizers with biostimulant properties on processing tomato crops (cv Taylor F1). This cultivar, provided by Nunhems Italia–BASF Electrical Seeds and belonging to the pear-shaped fruit type, is commonly grown in the agricultural areas in which the experiments took place. Prior to transplanting, a biodegradable black mulching film (15 µm thick MaterBi^®^, Novamont, Novara, Italy) was manually placed on each row.

The first trial (named “Trial A”) was conducted in 2021, in two separate farms: Croella (41°26′53.1″ N; 15°35′31.1″ E; 41 m a.s.l) (Field 1) and Palumbo (41°24′11.6″ N; 15°40′47.3″ E; 41 m a.s.l) (Field 2), approximately 3 km apart.

The second trial (named “Trial B”) was carried out in 2023, in the same Croella farm (Field 1). 

The soils in these farms are clayey vertisol of alluvial origin (1.20 m depth) (Typic Chromoxerert, fine, thermic, according to the Soil Taxonomy–USDA–NRCS [41]). Table 2 presents the main physico-chemical characteristics of surface soil, indicating differences between the two farms. The average percentage of sand was higher in Field 1 (36.8%) than in field 2 (18.9%), while clay and silt were higher in field 2 (45.1 and 36.0%, respectively) than in Field 1 (30.5 and 32.7%, respectively). Therefore, the textural class of the first soil was clay–loam, while the second soil was clay. Furthermore, although overall the chemical fertility of both soils is classified as well endowed, Field 1 had a slightly lower nutritional content (organic matter = 1.4%, N = 1.5‰, P_2_O_5_ = 56 mg kg^−1^, K_2_O = 1390 mg kg^−1^) and higher Ca (3128 mg kg^−1^) compared with field 2 (organic matter = 1.7%, N = 2.1‰, P_2_O_5_ = 75 mg kg^−1^, K_2_O = 1640 mg kg^−1^, Ca = 3008 mg kg^−1^).

In “Trial A”, tomato seedlings were transplanted on 20 April and harvested on 30 August in both fields. Radicon^®^ [42], a commercial organo-mineral fertilizer, was applied in each field by foliar spraying. This product is a non-microbial biostimulant that can be included in the class of humic and amino acid substances. The treatments were applied three times during the growing season (at transplantation, pre-flowering and enlargement of berry fruit development), compared with a control (no biostimulants added). The plants were sprayed until runoff, in conditions of low evapotranspiration demand (relative humidity: >70%; air temperature: <25 °C; wind speed: <8 km/h), typical of the early morning and late evening hours. This timing was chosen as the water in the solution in which the biostimulant is dissolved tends to evaporate slowly, thereby increasing the absorption of bioactive substances.

In “Trial B” tomato seedlings were transplanted at a later period than 2021 (on 20 May 2023) due to a prolonged period of rain that occurred in April and early May of this year (see Section 2.2), while the harvest took place on 12 September. Three commercial organo-mineral fertilizers were applied by fertigation as follows: Futuroot^®^ [Nutribiotech Srl] [43]; Radicon^®^ (Fertek) [42]; combinations of [Radicon^®^ + Amifort-Plus^®^] (Fertek). Similar to “Trial A”, these treatments were carried out in three waterings during the tomato crop cycle (at transplant, in pre-flowering and berry development), compared with the untreated control (Table 3).

In both trials, tomato seedlings (with 4–5 leaves and 100–150 mm height) were manually transplanted in the field in mulched paired rows (40 cm apart) spaced 180 cm apart, with the plants at a distance of 30 cm from each other along each single row, reaching a density of 3.7 plants m^−2^. Drip irrigation was used with lines between each pair of plant rows under the black plastic mulch. The crop was fully irrigated, meeting 100% of the crop evapotranspiration (ET_C_) calculated daily using the Doorembos and Pruitt equation [44]: ETc = EV × Kc_(EV)_, where ETc is the maximum daily crop evapotranspiration (mm), EV is the class A pan evaporation (mm), and Kc_(EV)_ is the crop coefficient referred to the class A evaporation pan, and the values of which depend on the growth stage of the plant (the values vary between 0.35 and 1.06). The Kc_(EV)_ values used in this study were 0.35 for the transplanting–rooting phase, 0.6 for the vegetative growth phase, 0.95 for the flowering phase, 1.07 for the fruit set, and 0.7 for the final phase [45]. Each irrigation was applied so as to maintain an interval of 3–4 days and was stopped 10 days before harvest. Applied irrigation volumes were measured using a water meter. Chemical fertilizers were used to meet the nutritional demands of crops based on the Apulia Region Fertilization Guide [46]. The nutrient needs were calculated: for pre-transplant fertilization, 35 kg ha^−1^ N and 70 kg ha^−1^ P_2_O_5_ were applied, while, during the tomato crop cycles, 75 kg ha^−1^ N and 100 kg ha^−1^ P_2_O_5_ were added by fertigation. Weed and pest control, and other agricultural managements employed were those commonly adopted by local farmers (weed control is carried out mechanically).

The experimental design in both “Trials” was a randomized block with three replications, and each treatment was compared in plots of 20 m^2^.

### 2.2. The Climate

The research site had a typical semi-arid area with a Mediterranean climate characterized as an accentuated thermo-Mediterranean climate [47], and further by the strong speed wind. The temperatures in this region can range from below 0 °C in winter to exceeding 40 °C in summer. The precipitation is unevenly distributed throughout the year, with the majority concentrated in the winter months, resulting in a long-term annual average of 559 mm [48]. Daily climate parameters, including maximum and minimum temperatures, wind speed and total precipitation, during the two growing seasons, were recorded from the weather station closest to the experimental area, provided by Syngenta [49], while evaporation from the class A pan evaporimeter (EV) was measured at the Croella farm. Table 4 outlines the climate conditions for the 2021 and 2023 seasons. The weather conditions between the two seasons were quite different. In fact, although the maximum and minimum temperatures for both seasons were similar (average of 29.9 °C and 14.3, respectively, in 2021, and 29.4 °C and 15.3, respectively, in 2023), in particular the month of July 2023, which saw an average maximum temperature that was much warmer (37.9 °C) compared with 2021 (35.4 °C). The average daily wind speed varied between 3.3 and 4.3 m s^−1^. Furthermore, in 2023, prolonged rainfall was recorded in the months of April and early May (66.2 and 73.6 mm, respectively) which, as previously mentioned, delayed the transplanting of tomato seedlings (Table 4).

### 2.3. Leaf Green Color Intensity (SPAD)

In “Trial A”, the leaf green colour intensity (SPAD) was measured during the reproductive phase of the tomato on three days: 1, 8 and 28 June 2021 (a respective 42, 49 and 69 days after transplanting (DAT) in Field 1, and 28, 35, and 55 DAT in Field 2). The SPAD, an indirect measurement index of chlorophyll, was taken from the youngest fully expanded leaves, using a SPAD-502 portable chlorophyll meter (Konica-Minolta corporation, Ltd., Osaka, Japan). Ten leaves per plot were randomly measured to obtain an average of 20 readings (10 on each side of the leaf) per plot.

### 2.4. Harvesting, Fruit Collection, Yield and Fruit Qualitative Analysis

At tomato harvest, on a 6 m^2^ sampling area, the following yield components were evaluated: marketable yield (t ha^−1^), unmarketable yield (t ha^−1^) (green fruits and rotten fruits) and green plant biomass (t ha^−1^). On a sample of 10 marketable fruits from each plot, the following quality parameters were measured: mean fresh weight (W_m_; g), fruit length (L; mm), fruit width (W; mm), dry-matter content (DM; % fruit fresh matter); soluble solids content (SSC; °Brix), pH, titratable acidity (TA; g citric acid per 100 mL^−1^ fresh juice) [50] and a*/b* ratio (CI) [51]. 

The colour parameters were measured using a spectrophotometer (CM-700d; Minolta Camera Co., Ltd., Osaka, Japan) as the CIELAB coordinates (i.e., L*, a*, b*) on four randomly selected areas of the fruit surface. Only the a*/b* ratio has been reported, which represents an index that describes the colour differences of tomato fruit [52,53]. The total phenol content was determined according to the method of Singleton and Rossi [54] and expressed as mg gallic acid equivalents (mg GAE/g dw). Lycopene content was analysed by HPLC on a reversed-phase C30 column and binary gradient made of a methanol/water solution and dichloromethane [55] and expressed as mg/100 g fw.

### 2.5. Statistical Analysis

The measured data from SPAD and from each of the continuum variables relating to the qualitative/quantitative traits of the tomato fruit were processed using analysis of variance (ANOVA) performed with JMP^®^ software version 8 (SAS Institute Inc., Cary, NC, USA). Average values were compared using Tukey’s test with a degree of confidence of 95% (*p* < 0.05). Standard deviations (SD) were calculated using Excel software from the Office 2007^®^ suite (Microsoft Corporation, Redmond, WA, USA). Percentage values were transformed to arcsine before conducting the analysis of variance.

## 3. Results

### 3.1. Trial A: Fields 1 and 2

#### 3.1.1. SPAD Index, Yield and Fruit Quality

SPAD readings over three days during the tomato crop cycle (Figure 1) were significantly higher for plants supplied with Radicon^®^ (average 55.3 in Field 1 and 63.8 in Field 2) than those of the control (average 49.8 in Field 1, and 58.0 in Field 2).

Furthermore, as seen in Figure 2, a linear relationship was found between the SPAD index and marketable yield (R = 0.726), where the increase in yield was directly related to the increase in SPAD values.

The impact of the biostimulant Radicon^®^ on the quantitative and qualitative components of the tomato crops’ yield is shown in Figure 3. In both Fields 1 and 2, significantly higher marketable yields were observed with the biostimulant treatment (averaging 119.4 and 129.1 t ha^−1^, respectively) compared with the controls (averaging 102.8 and 120.8 t ha^−1^, respectively). Thus, foliar application of Radicon^®^ increased marketable yield by 16.1% in Field 1 and 6.8% in Field 2 compared with controls. However, the marketable yield, on average, was greater in the fertile Field 2 (125.0 t ha^−1^) than in Field 1 (111.1 t ha^−1^).

For both green and rotten fruits, no significant differences were found between the treatment and control in both Fields 1 and 2. However, the average values were significantly higher in Field 2 (8.7 and 10.1 t ha^−1^, respectively) compared with Field 1 (7.9 and 9.1 t ha^−1^ respectively.)

Similar differences were observed for the total fruit yield, with significantly higher yields in both Fields 1 and 2 with Radicon^®^ treatment (on average 137.0 and 149.2 t ha^−1^, respectively) than the control (averaging 119.2 and 138.1 t ha^−1^, respectively).

No significant differences were found in plant biomass between the biostimulant treatment and control in both Fields 1 and 2 (on average 22.2 t ha^−1^), but higher mean values were observed in Field 2 (averaging 24.8 t ha^−1^) compared with Field 1 (averaging 19.7 t ha^−1^).

Regarding the weight, length and width of fruits, statistical differences were noted neither between the biostimulant treatment and control, nor between fields (on averaging 77.7 g, 76.6 mm, 41.4 mm, respectively). Notably, fruit weight in both fields was slightly higher in the treated crop (on averaging 79.4 g) compared with the control (averaging 76.5 g) (Figure 3).

#### 3.1.2. Physico-Chemical Parameters of the Fruit

Table 5 shows the mean values for the qualitative physico-chemical characteristics of the processing tomato fruit concerning the Radicon^®^ treatment and the control at the two separate fields.

The fruit parameters—SSC, TA, pH, DM, total phenols, lycopene and colour index (CI)—showed no statistically significant differences between either of the two biostimulant treatments and control nor between fields. The mean values varied between 4.1 and 4.4 °Brix for SSC, 0.39 and 0.41 g citric acid 100 mL^−1^ juice for TA, 4.21 and 4.34 for pH, 5.5 and 6.1% for DM, 2.21 and 2.30 mg GAE/g dw for total phenols, 0.80 and 0.91 mg/g fw for lycopene, and 0.95 and 1.18 for the colour index (a*/b* ratio).

However, it should be noted that the values for DM and lycopene tended to be higher with Radicon^®^ treatment (averaging 6.0% and 0.90 mg/g fw) compared with control (averaging 5.5% and 0.83 mg/g fw).

### 3.2. Trial B (2023 Season)

#### 3.2.1. Yield and Fruit Quality

Figure 4 illustrates the impact of biostimulant treatments with Futuroot^®^, Radicon^®^ and the combination of [Radicon^®^ + Amifort^®^] on quantitative and qualitative components of yield. The average marketable yield showed significantly higher values for both Futuroot^®^ (117.1 t ha^−1^) and [Radicon^®^ + Amifort^®^] (116.2 t ha^−1^) compared with Radicon^®^ (104.0 t ha^−1^), which, in turn, was significantly higher than the control (96, 9 t ha^−1^). Thus, soil application of Futuroot^®^, [Radicon^®^ + Amifort^®^], and Radicon^®^ increased the average marketable yield by 27.8%, 27.7% and 13.5%, respectively, over the controls.

Green fruits at harvest were significantly higher in the Futuroot^®^ (2.9 t ha^−1^) compared with both Radicon^®^ (1.7 t ha^−1^) and [Radicon^®^ + Amifort^®^] (1.8 t ha^−1^) treatments, which, in turn, were significantly higher than the control (on average 1.1 t ha^−1^). Furthermore, rotten fruits were significantly higher in the [Radicon^®^ + Amifort^®^] (8.1 t ha^−1^) than Radicon^®^ (7.2 t ha^−1^), which, in turn, were significantly higher than both Futuroot^®^ (3.8 t ha^−1^) and the control (4.2 t ha^−1^). Similar to the marketable yield, the total fruit yields obtained in both Futuroot^®^ (123.3 t ha^−1^) and [Radicon^®^ + Amifort^®^] (128.1 t ha^−1^) were significantly higher than Radicon^®^ (112.9 t ha^−1^), which, in the shift, was significantly higher than the control (112.6 t ha^−1^). Furthermore, plant biomass was also significantly higher in the [Radicon^®^ + Amifort^®^] (15.9 t ha^−1^), Radicon^®^ (13.9 t ha^−1^), and Futuroot^®^ (t 15.0 t ha^−1^) treatments than the control (12.1 t ha^−1^).

Regarding the average weight of the fruits, although no difference was noted between the treatments (on average 68.1 g), they were significantly higher than the control (65.0 g).

Finally, concerning the length and width of the fruits, slightly higher values were detected for the biostimulant treatments (averaging 83.1 and 41.4 mm, respectively) compared with the control (averaging 78.1 and 38.5 mm, respectively) (Figure 4).

As regards the plant biomass, all biostimulant treatments provided values significantly higher (on average 15.0 t ha^−1^) than the control (13.7 t ha^−1^).

#### 3.2.2. Physico-Chemical Parameters of the Fruit

None of the fruit physico-chemical parameters—SSC, TA, pH, DM, total phenols, lycopene and colour index (CI)—showed statistically significant differences between biostimulant treatments and control. The mean values varied between 4.90 and 5.17 °Brix for SSC, 0.30 and 0.31 g citric acid 100 mL^−1^ juice for TA, 4.66 and 4.74 for pH, 5.63 and 6.36% for DM, 2.30 and 2.41 mg GAE/g dw for total phenols, 0.80 and 0.91 mg/g fw for lycopene, and 1.03 and 1.17, for the colour index (a*/b*ratio) (Table 6).

## 4. Discussion

To increase agriculture production, the application of chemical fertilisers is indispensable. However, intensive cultivation and monoculture have contributed to the widespread use of chemical fertilizers, leading to a decline in organic soil fertility [56]. This negative effect can be reduced by adopting new agricultural technological practices, including the application of soil biostimulants that improve soil organic matter and nutrient content [57].

The present study, consisting of two independent trials conducted in two different years (2021 and 2023), reports the results of three foliar or soil applications of different organo-mineral fertilizers with biostimulant action, given during the crop cycle, on the yield and quality of processing tomato (cv Taylor F1).

In Trial A (2021 season), the foliar application of Radicon^®^, containing humic and fulvic acids, was investigated on tomatoes grown in two soils with different physical-chemical fertility (Field 1: less fertile; Field 2: fertile).

The results show that the plants sprayed with Radicon^®^ significantly increased the relative chlorophyll content, as measured by SPAD testing, in tomato leaves during plant development, compared with an untreated control. This result was similar to those obtained in a previous study on tomato in which a biostimulant composed of algae, fish and humus extracts [15] or two Chlorophyta microalgae [58] were used, but was different from another study [59] in which no effect was noted using two formulations obtained from A. nodosum. The SPAD assessment serves as a reliable indicator of photosynthetic activity, leaf nitrogen status and leaf chlorophyll content [60,61]; therefore, higher SPAD values reflect better plant health [62]. A linear positive relationship was found between the SPAD index and marketable yield (R = 0.726). This result is in line with that obtained by Nemeskéri et al., 2019 [63]. In this regard, our SPAD values are higher in plants grown in the field characterized by better fertility.

Plants treated with Radicon significantly increased tomato productivity in both soil conditions. These results are consistent with those recorded in previous studies [37,64,65] in which biostimulants based on humic acids were used on plants. Furthermore, the statistical analysis of the results reveal that, although the marketable yield, as a field average, was greater in the fertile soil (125.0 t ha^−1^) than in the less fertile one (111.1 t ha^−1^), significant increases were greater in less fertile soil (16.1%) than in fertile ones (6.8%). These results are in agreement with those recorded in previous studies [17,65,66]. In general, the yield increases obtained with the Radicon can be explained by the heavier fruit (79.4 g), although with values slightly higher than the control (76.5 g).

No relevant effects of Radicon^®^ treatment on some quality attributes of tomato fruits (SSC, TA, pH, total phenols and CI) were observed. These results are in agreement with those found in previous research [64], but are in disagreement with others [32,67] in which the same parameters were improved by biostimulants.

As for DM and lycopene content, the average values tended to be higher in the biostimulant treatment (average 6.0% and 0.90 mg/g fw) compared with control (average 5.5% and 0.83 mg/g dw). In this regard it is known that high values of these parameters are also desirable characteristics for the canned tomato industry as it improves the quality of the processed product [32,53,68,69]. It is clear that the foliar application of Radicon^®^ in tomato crop may be a practical option for achieving higher marketable yields and some quality attributes of tomato fruits, increasing the market value of the crop.

In Trial B (2023 season), the soil application of Futuroot^®^, Radicon^®^, and the combination of [Radicon^®^ + Amifort-Plus^®^] in the same less fertile soil as that used in the first trial, significantly increased the marketable yield by 27.8%, 13.5% and 27.7%, respectively, compared with the control. Similar increases were obtained in total fruit yield by 27.2%, 16.5% and 28.1%, respectively, compared with the control. In general, these yield increases obtained in crops treated with biostimulants can be explained by the heavier fruits (on average 68.1 g) and/or the higher plant biomass (on average 15.0 t ha^−1^) compared with the control (on average, 65.0 g and (13.7 t ha^−1^, respectively).

Our yield increases are consistent with those recorded in previous studies [34,38,70] in which humic or protein hydrolysate-based biostimulant were added to the soil.

The best beneficial effect ascribed in the case of Futuroot^®^ could be due not only to the humic acids but also to the auxins, cytokinins and microelements (Zn and Mn) present in it, and in the case of [Radicon^®^ + Amifort Plus^®^] to the combined distribution of the humic acids present in Radicon^®^ and the magnesium and n-chelated microelements (MgO) present in Amifort Plus^®^, which improved plant productivity. These results are in agreement with those found in previous research [71,72] where the application of Zn and chelated trace elements (MgO) significantly increased fruit yield in tomato. Furthermore, in the crop treated with Futuroot^®^ the greatest weight of green fruits was recorded at harvest, indicating that not all the fruits of the last bunches were yet ripe on the plants slightly late in the ripening phase.

Finally, in this trial the physico-chemical parameters of the fruit (SSC, TA, pH, DM, total phenols, lycopene and CI) were statistically different neither among the biostimulant treatments nor between these and the control. In this regard, it is highlighted that the effects of different biostimulants on tomato quality obtained in the literature are very different from each other. These differences could be due to several factors, among which are the biostimulant composition [35] and the technique of their application (when and how to apply) [73,74], the tomato genotype used [17,70] and the climatic conditions, such as temperature [75].

From the comparison between the results obtained in Trials 1 and 2, it is highlighted that, in general, the final yield of the tomato, although at different levels, showed an increase with all of the biostimulant formulations used, when applied three times during the crop cycle both to the leaves and in the soil, when compared with the untreated control. Furthermore, it should be noted that, when Radicon^®^ was applied alone in both experiments on the same Field 1, increases in marketable yield compared with the control were slightly higher with foliar application (16.1%) than with the soil application (13.5%). However, the soil application of Futuroot^®^ or [Radicon^®^ + Amifort-Plus^®^], containing not only biostimulant compounds but also microelements (Z, Mg, Mn), provided the best tomato yield.

Regarding the qualitative aspects, the following can be stated. Our data of DM (varied between 5.5% and 6.4%) SSS (varied between, and 4.3 °Brix and 5.2 °Brix) are in agreement with [76], which reports values ranging from 4 to 7% and from 3.3 °Brix to 5.5 °Brix, respectively, in a wide field experiment carried out in the Mediterranean basin aiming to compare the quality characteristics of 33 tomato genotypes. The average TA data varied between 0.39 and 0.45%, which is very close to the average acidity of processing tomatoes, which is generally around 0.35 [77]. The pH range values (4.21–4.74) were within the typical range of tomato fruits [78] and showed their slightly acidic nature.

The CI, which is commonly used as a quality redness index (brightness of red colour) of tomatoes was found in the range from 0.95 to 1.18, in agreement with the range from 0.95 to 1.21 observed by [79] on tomato fruits at the red stage of maturity.

Finally, as reported by Riga et al. [80], tomato quality is more dependent on temperature than on photosynthetically active radiation.

## 5. Conclusions

It should be considered that, unless part of the research concerns the comparison between the treatment with Radicon^®^ conducted in the same field, it is not possible to draw unifying conclusions between the two trials conducted in different sites. However, for each of them the results provided very interesting indications. In general, biostimulant treatments through application of the Radicon^®^ product by foliar or by fertigation increased the yield when compared with the untreated control, while slight effects were found on some qualitative attributes of tomato fruits. Furthermore, through the foliar application of the Radicon^®^ product, the best production results were obtained in low fertility soil conditions and there was a significant increase in the total chlorophyll pigment content compared with the control. The soil application of Futuroot^®^ or the combination [Radicon^®^ + Amifort-Plus^®^) products containing not only biostimulant compounds but also auxins, cytokinins and microelements (Z, Mg, Mn), also increased yield compared with the untreated control, while slight effects were found on some qualitative attributes of tomato fruits. Furthermore, when also considering the higher production costs, Futuroot^®^ or the combination of [Radicon^®^ + Amifort-Plus^®^] applied to soil by fertigation are considered particularly optimal for regular use in conventional agricultural practices of processing tomato crop. Ultimately, the findings of this study, in terms of yield mentioned above, can be helpful to assist growers in the practice of fertilizing tomato crop and ensuring product quality for industrialists.

## Figures and Tables

**Figure 1 plants-13-01458-f001:**
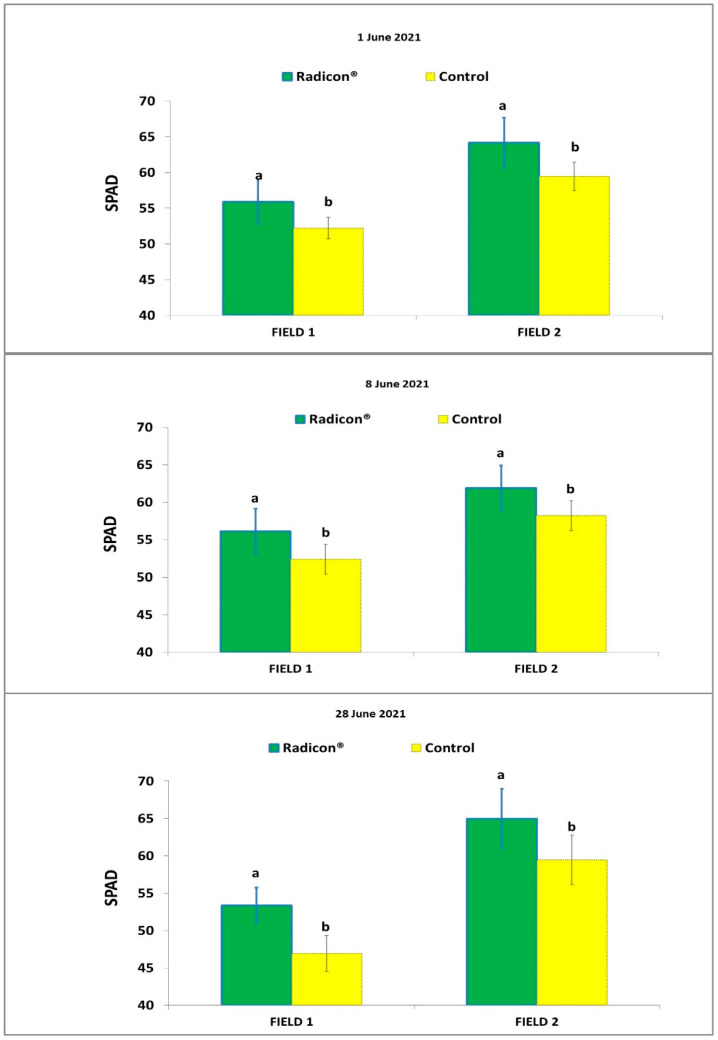
Average SPAD values ± std. dev. detected on 1, 8 and 28 June 2021, on tomato plants treated with biostimulant and the untreated control grown in Fields 1 and 2. Different letters indicate significant differences at *p* < 0.05, according to Tukey’s test.

**Figure 2 plants-13-01458-f002:**
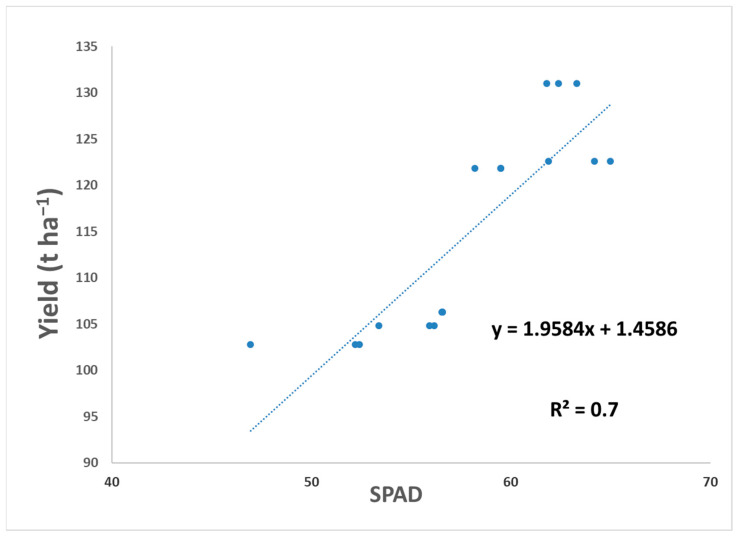
Linear SPAD index—marketable yield ratio obtained for biostimulant-treated and untreated control tomato crops grown in both Fields 1 and 2.

**Figure 3 plants-13-01458-f003:**
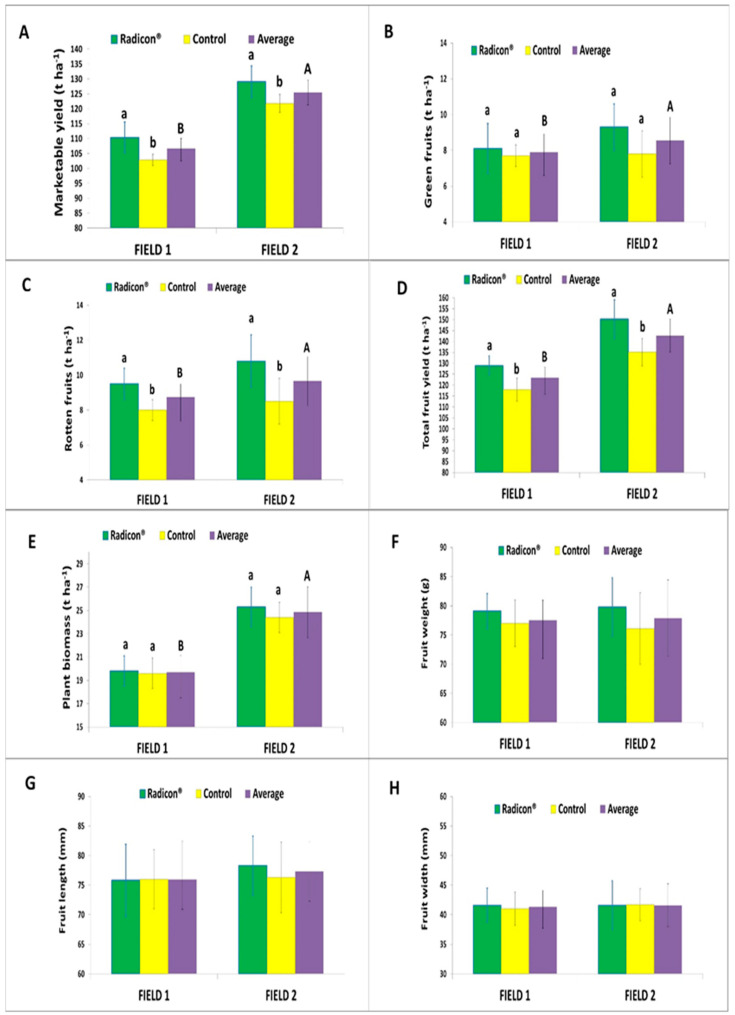
Effect of biostimulant treatment and control on (**A**) marketable yield, (**B**) green fruits, (**C**) rotten fruits, (**D**) total yield, (**E**) plant biomass, (**F**) fruit weight, (**G**) fruit length, and (**H**) fruit width. Average values ± std. dev. of biostimulant treatment and control in each field and the relative average of two fields are shown (different lowercase letters per field and different capital letters between fields each indicate significant differences at *p* < 0.05). Graph without letters means there were no significant differences between mean.

**Figure 4 plants-13-01458-f004:**
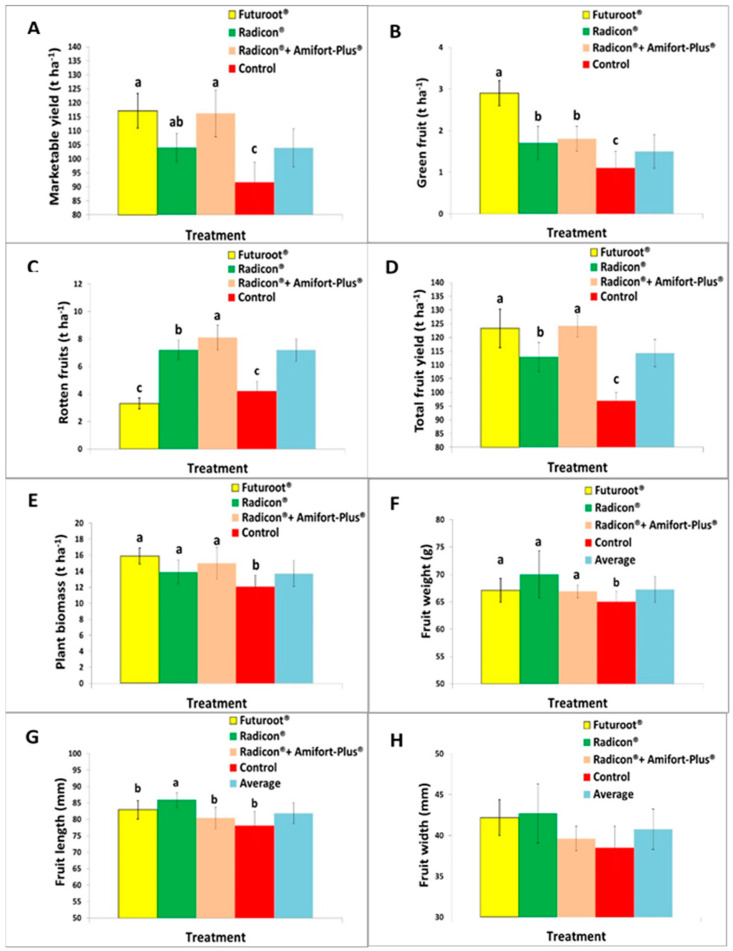
Effect of biostimulant treatments and control on (**A**) marketable yield, (**B**) green fruits, (**C**) rotten fruits, (**D**) total yield, (**E**) plant biomass, (**F**) fruit weight, (**G**) fruit length, and (**H**) fruit width. Average values ± std. dev. of biostimulant treatments and control are shown (different lowercase letters indicate significant differences at *p* < 0.05). Graph without letters means there were no significant differences between mean.

**Table 1 plants-13-01458-t001:** Articulation of the research.

Trial Season	Field	Biostimulant Treatments	Mode of Application
Trial A 2021	1	Radicon^®^ vs. Control	Foliar application
	2	Radicon^®^ vs. Control	Foliar application
Trial B 2023	1	Futuroot^®^	Soil drench by fertigation
		vs.	Soil drench by fertigation
		Radicon^®^	
		vs.	Soil drench by fertigation
		[Radicon^®^ + Amifort-Plus^®^]	
		vs.	
		Control	

**Table 2 plants-13-01458-t002:** Main physico-chemical characteristics of the soils at the two experimental fields.

Characteristics		Field 1	Field 2
		Croella Farm	Palumbo Farm
Sand [2.0 > Ø < 0.02 mm]	(%)	36.8	18.9
Loam [0.02 > Ø < 0.002 mm]	(%)	32.7	36.0
Clay [Ø < 0.002 mm]	(%)	30.5	45.1
Typology (USDA)		Clay–loam	Clay
Organic matter (Walkley–Black)	(%)	1.4	1.7
pH (soil:water 1:2.5)		8.0	8.0
ECe on 1:2 (*w*/*v*) (aqueous soil extract)	(dS cm^−1^)	0.7	0.5
Total nitrogen (Kjeldhal)	(mg kg^−1^)	1.5	2.1
P_2_O_5_ available (Olsen)	(mg kg^−1^)	56	75
K_2_O exchangeable (Shollemberger)	(mg kg^−1^)	1390	1640
Ca exchangeable	(mg kg^−1^)	3128	3008

The “Ø” denotes diameter; *w*/*v* means weight/volume.

**Table 3 plants-13-01458-t003:** Formulations and doses of foliar e/o fertigation application of agricultural biostimulant products used in the experiments.

Organo-Mineral Fertilizer Treatment
**Futuroot^®^** (Nutribiotech): based on zinc, auxins, cytokinins and plant amino acids. Composition: Mn, soluble in water 2%, zinc sulphate soluble in water 9%. Activated with plant amino acids and humic acids. Applied by Fertigation: 12 kg ha^−1^.
**Amifort-Plus^®^** (Fertek): organic nitrogen fertilizer fluid flesh in suspension with magnesium and chelated microelements N (MgO) + C 6 (2) + 14, obtained from the hydrolysis of animal protein substances. Applied by Fertigation: 2 kg per 1000 m^−2^.
**Radicon^®^** (Fertek): a suspension–solution of humic and fulvic acids, obtained from worm compost (night crawled). Dry composition: total organic matter 60%; extractable organic substance 4% of organic matter; humified organic substance 90% extractable organic matter; organic substance 1.0% of extractable organic nitrogen; C/N ratio = 4; Fertigation: 2 kg per 1000 m^2^. Foliar fertilization applied at a dose of 500 g 100 L^−1^ of water.

**Table 4 plants-13-01458-t004:** Average monthly maximum and minimum temperatures (T_max_, T_min_), wind speed (Ws), total precipitation (P), total evaporation from the class A pan evaporimeter (EV) and average crop coefficient referring to the class A pan evaporation (Kc_(EV)_) for the 2021 and 2023 seasons.

Month	T_max_	T_min_	W_s_	P	EV	Kc
	(°C)	(°C)	(m s^−1^)	(mm)	(mm)	(EV)
			2021			
April	19.9	4.7	4.3	40.4	161.3	0.35
May	26.5	10.8	3.5	26.0	203.2	0.50
June	33.2	15.9	3.3	8.6	230.2	0.95
July	35.4	19.3	3.7	100.8	247.8	1.03
August	34.9	19.4	3.8	29.2	238.9	0.80
September	29.5	15.4	3.5	19.4	192.4	----
Mean	29.9	14.3	3.7			0.73
Total				224.4	1273.8	
			2023			
April	18.0	8.1	3.5	66.2	162.8	----
May	24.0	11.9	3.5	73.6	200.2	0.40
June	31.1	16.0	3.8	84.0	225.9	0.78
July	37.9	19.9	3.5	3.2	255.5	1.06
August	34.6	18.4	4.2	28.4	218.0	0.76
September	30.6	17.4	4.0	17.6	204.0	0.60
Mean	29.4	15.3	3.8			0.72
Total				273.0	1266.4	

**Table 5 plants-13-01458-t005:** Average values for the fruit chemical–physical parameters of the tomato crops treated with biostimulant by foliar application and the control at Fields 1 and 2 (“Trial A”, 2021 season).

Parameters Evaluated	Field	Treatment		Average
		Radicon^®^	Control	
SSC (°Brix)	1	4.4 ± 0.4	4.3 ± 0.3	4.3 ± 0.3
	2	4.3 ± 0.3	4.2 ± 0.3	4.3 ± 0.3
TA (g citric acid 100 mL juice)	1	0.39 ± 0.2	0.41 ± 0.1	0.40 ± 0.1
	2	0.41 ± 0.1	0.40 ± 0.2	0.41 ± 0.1
pH	1	4.21 ± 0.06	4.24 ± 0.06	4.23 ± 0.06
	2	4.30 ± 0.03	4.34 ± 0.05	4.32 ± 0.04
DM (%)	1	6.0 ± 0.4	5.5 ± 0.4	5.7 ± 0.4
	2	6.1 ± 0.7	5.5 ± 0.5	5.8 ± 0.6
Total Phenols (mg GAE/g dw)	1	2.21 ± 0.1	2.23 ± 0.1	2.22 ± 0.1
	2	2.30 ± 0.2	2.26 ± 0.1	2.28 ± 0.1
Lycopene (mg/100 g fw)	1	0.89 ± 0.06	0.80 ± 0.10	0.84 ± 0.08
	2	0.91 ± 0.09	0.86 ± 0.06	0.90 ± 0.07
Colour index (a*/b* ratio)	1	1.02 ± 0.07	0.95 ± 0.09	0.98 ± 0.08
	2	1.16 ± 0.10	1.18 ± 0.21	1.17 ± 0.11

The data are averages ± std. dev. of biostimulant treatment and control in each field and the relative average of two fields. The absence of letters indicates no significant differences among both treatments and fields.

**Table 6 plants-13-01458-t006:** Average values for the fruit physico-chemical parameters of the tomato crop treated with biostimulants applied by fertigation and the control (Trial B, 2023 season).

Parameters Evaluated	Treatment				Average
	Futuroot^®^	Radicon^®^	Radicon^®^ + Amifort-Plus^®^	Control	
Soluble solids content (°Brix)	4.97 ± 0.25	5.17 ± 0.31	5.10 ± 0.20	4.90 ± 0.26	5.06 ± 0.25
TA (g citric acid 100 mL juice)	0.30 ± 0.03	0.30 ± 0.03	0.31 ± 0.03	0.31 ± 0.04	0.30 ± 0.03
pH	4.74 ± 0.04	4.66 ± 0.03	4.70 ± 0.04	4.70 ± 0.03	4.69 ± 0.04
Dry matter (%)	6.35 ± 0.24	6.00 ± 0.08	5.63 ± 0.25	6.36 ± 0.23	6.00 ± 0.19
Phenols (mg GAE/g fw)	2.33 ± 0.09	2.41 ± 1.00	2.30 ± 0.08	2.31 ± 0.08	2.34 ± 0.31
Lycopene (mg/g dw)	0.94 ± 0.05	0.96 ± 0.03	0.95 ± 0.04	0.92 ± 0.03	3.1 ± 0.04
Colour index (a*/b* ratio)	1.03 ± 0.12	1.06 ± 0.10	1.17 ± 0.07	1.06 ± 0.09	1.08 ± 0.09

Data are means ± std. dev. The absence of letters indicates no significant differences among both treatments and fields.

## Data Availability

Data are contained within the article.

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
