# Peer review of "Yield and Fruit Characteristics of Tomato Crops Grown with Mineral Macronutrients: Impact of Organo-Mineral Fertilizers through Foliar or Soil Applications"

_plants, 2024, doi:10.3390/plants13111458_

Round 1
Reviewer 1 Report
Comments and Suggestions for Authors
The study used two experiments to evaluate the effects of Organo-mineral fertilizers through foliar or soil applications on yield and fruit characteristics of tomato. Although the research content has some practical value and significance, the specific two experiments can not be seen to have internal links (Figure 1). In addition, the processing of the study settings was relatively simple, and the results were not analyzed in depth.
There are many problems in the writing of the manuscript. There are many wrong expressions and misspellings. For example, line 42, Consguently should be Consquently.
According to the presentation form and writing level of the research content of the manuscript, it is far from the level of publication.
Comments on the Quality of English LanguageThere are many problems in the writing of the manuscript. There are many wrong expressions and misspellings. For example, line 42, Consguently should be Consquently.
Author Response
Dear Reviewer,Thanking you for your comments to which we respond as follows:
1) The reviewer wrote: The results were not analyzed in depth.
Our answer: The results were reanalyzed in depth, several sentences were changed or eliminated or added in order to improve the text. All corrections are indicated in red in the text.
Furthermore, new recent bibliographic references have been added:
Leogrande, R.; Lopedota, O.; Vitti, C.; Ventrella, D.; Montemurro, F. Saline water and municipal solid waste compost application on tomato crop: effects on plant and soil. J Plant Nutr Soil Sc. 2016, 39(4), 491-501 doi: 10.1080/01904167.2015.1084325.
FAOSTATS 2022. Available online: https://www.fao.org/faostat/en/#search/almond (accessed on 23 March, 2024).
Rouphael, Y., and Colla, G. Biostimulants in agriculture. Front. Plant Sci. 2020, 11, 40. doi: 10.3389/fpls.2020.00040.
Weisser, M.; Mattner, S.W.; Southam-Rogers, L.; Hepworth, G.; Arioli, T. Effect of a fortified biostimulant extract on tomato plant productivity, physiology, and growing media properties. Plants 2024, 13, 4. https://doi.org/ 10.3390/plants13010004.
Hellequin, E.; Monard, C.; Chorin, M.; Daburon, V.; Klarzynski, O.; Binet, F. Responses of active soil microorganisms facing to a soil biostimulant input compared to plant legacy effects. Sci. Rep. 2020, 10, 1–15.
Della Lucia, M.C.; Bertoldo, G.; Broccanello, C.; Maretto, L.; Ravi, S.; Marinello, F.; et al. Novel effects of leonardite-based applications on sugar beet. Front. Plant Sci. 2021, 12:646025. doi: 10.3389/fpls.2021.646025.
Distefano M.; Mauro, RP.; Page D.; Giuffrida, F.; Bertin, N.; Leonardi, C. Aroma volatiles in tomato fruits: the role of genetic, preharvest and postharvest factors. Agronomy. 2022. 12, 376. doi:10.3390/agronomy12020376.
Riga, P.; Anza, M.; Garbisu, C. Tomato quality is more dependent on temperature than on photosynthetically active radiation. J. Sci. Food Agric. 2008, 88, 158–166. doi: 10.1002/jsfa.3065
2) The reviewer wrote: There are many problems in the writing of the manuscript. There are many wrong expressions and
Our answer: the manuscript was revised and the English language corrected.
We send you sincere greeting
Grazia Disciglio, Annalisa Tarantino and Laura Frabboni
Reviewer 2 Report
Comments and Suggestions for Authors
First of all, I appreciate this work. It's already presented in a good way but needs some changes before acceptance.
Here please find my recommendations:
Please start the abstract with a formal introduction about your topic.
Line 13: Please change "These applications were made" to The treatments were applied"
Please mention the increased parameters of both experiments and also write which one is better.
Line 33: Please remove the keywords that are already used in the title. Keywords should be different.
Line 37-39: In the province of Foggia, the cultivation of processing tomatoes (Lycopersicon escu-37 lentum Mill.) holds the Italian record, covering an area of over 15,000 hectares and boast-38 ing production of approximately 1.425 million tons. Please start with the worldwide importance of tomato production then specifically on your region.
Line 66-69:"The results of a study by Caradonia et al. [25] on the applications of different micro-66 bial biostimulants on processing tomato reported positive influences on morphological 67 and physiological parameters when cultivated in a greenhouse. However, these effects were 68 reduced when cultivated in the open field due to environmental factors." In this paragraph, the author tried to correlate their study with existing literature. But this literature is not enough. A single-person study can not lead to conduct another experiment. Please relate your study with more literature.
Line 81: "The research was structured as shown in Figure 1." it is better to represent the experimental design in table form.
Line 444_445: From this research, carried out in two open-field separate experiments, it arose that 444 the biostimulant treatments of tomato crop using different organo-mineral fertilizers. Please be clear about what treatments are applied.
Also, a conclusion should be short.
Comments on the Quality of English Language
Extensive English editing is needed.
Author Response
Dear Reviewer, Thanking you for your appreciation of the manuscript and for the comments to which we respond and insert new sentences as follows:
The reviewer wrote: Please start the abstract with a formal introduction about your topic Our answer: The following sentence was introduced at the beginning of the abstract“The utilization of natural fertilizers has gained importance as a strategy to plant productivity and soil health”. Two independent trials were conducted in two seasons (2021 and 2023) to evaluate the effect of the foliar or soil application of different commercial organo-mineral fertilizers”
The reviewer wrote: Line 13: Please change "These applications were made" to The treatments were applied
Our answer: We changed the word as requested
The reviewer wrote: Please mention the increased parameters of both experiments and also write which one is better.
Our answer The following sentences was inserted in the “Abstract”:“Furthermore, the increase in marketable yield obtained with Radicon® applied to leaves was higher (16.1%) than that applied to soil (13.5%)”
The reviewer wrote: Line 33: Please remove the keywords that are already used in the title. Keywords should be different
Our answer: “Foliar spray and soil application” have been eliminated from the keywords.
The reviewer wrote: Line 37-39: In the province of Foggia, the cultivation of processing tomatoes (Lycopersicon escu-37 lentum Mill.) holds the Italian record, covering an area of over 15,000 hectares and boast-38 ing production of approximately 1.425 million tons. Please start with the worldwide importance of tomato production then specifically on your region.
Our answer: The following sentences was inserted at the beginning of the “Introduction”, and FAOSTAT ,2022 referencewas added:
“Tomato (Lycopersicon esculentum Mill.) is the vegetable crop with the highest demand and greatest economic value in the world. Therefore it is one of the most cultivated horticultural crops in the world (Leogrande et al., 2012). In 2022, the world annual tomato yield was 186.821 million metric tonnes over a cultivated area of 4,917,735 hectares (FAOSTAT,2022)
FAOSTATS 2022. Available online: https://www.fao.org/faostat/en/#search/almond (accessed on 23 March, 2024).
The reviewer wrote: Line 66-69:"The results of a study by Caradonia et al. [25] on the applications of different microbial biostimulants on processing tomato reported positive influences on morphological and physiological parameters when cultivated in a greenhouse. However, these effects were reduced when cultivated in the open field due to environmental factors." In this paragraph, the author tried to correlate their study with existing literature. But this literature is not enough. A single-person study can not lead to conduct another experiment. Please relate your study with more literature.
Our answerThe above sentence was poorly described; therefore, it has been changed as follows:
“Numerous research conducted on tomato crops in controlled environment or pods [13, 14, 7, 15-25], reported positive influenced on yield, morphological and physiological parameters. While reports on the potential of biostimulants in the field are less explored. This is mainly due to the variety of underlying factors in crop fields, including weather variability, climate fluctuations, soil type, and field management [7].
The reviewer wrote: Line 81: "The research was structured as shown in Figure 1." it is better to represent the experimental design in table form
Our answer: We have replaced figure 1 in table 1
The reviewer wrote: Line 444_445: From this research, carried out in two open-field separate experiments, it arose that the biostimulant treatments of tomato crop using different organo-mineral fertilizers. Please be clear about what treatments are applied.
Our answer: We have entered the names of the biostimulants used as follows:
“From this research, carried out in two open-field separate experiments, it arose that the biostimulant treatments of tomato crop using different organo-mineral fertilizers (Futuroot®, Radicon® and the combinations [Radicon® + Amifort-Plus®], …
The reviewer wrote: Also, a conclusion should be short.
Our answer: The conclusion has been slightly changed
We send you sincere greeting
Grazia Disciglio, Annalisa Tarantino and Laura Frabboni
Reviewer 3 Report
Comments and Suggestions for Authors
This work is novel, however need major improvements in the text and Data presentation in Tables.
1. Must start your abstract with a scientific problem not with your experiment explaination.
2. Hypothesis is not clear.
3. Please re-write your intorduction with latest references and avoid short paragraphs.
4. outline 2.4 need to change into (Furit harvesting and qualitative analysis.
5. Please must revise all tables, not in proper formate.
6. I suggest change Table 3 into Figure. Readers will easily understand.
7. Experiment 1 and experiment 2 is not a proper and scientific wording. I suggest to change it.
8. Also discussion need improvement, I suggest to add sub-headings, and need to add more explaination for the support of your results.
9. I suggest to add the results of correlation analysis.
10. Conclusion is well drawn but may i know, how your work is important for a local farmer, and for an industralist?
Author Response
Dear Reviewer,Thanking you for your appreciation of the manuscript and for the comments to which we respond and insert new sentences as follows:
1) The reviewer wrote: Must start your abstract with a scientific problem not with your experiment explaination.
Our answer: The following sentence was introduced at the beginning of the abstract“The utilization of natural fertilizers has gained importance as a strategy to plant productivity and soil health”
2) The reviewer wrote: Hypothesis is not clear.
Our answer: The following sentences were introduced:
The utilization of natural fertilizers has gained importance as a strategy to plant productivity and soil health” In this study we investigated the potential beneficial effect of the foliar or soil application of different commercial organo-mineral fertilizers with biostimulant action on the yield and fruit characteristics of processing tomato crops (cv Taylor F1) exposed to conventional macronutrients
3) The reviewer wrote: Please re-write your intorduction with latest references and avoid short paragraphs.
Our answer:The introduction was re-vrite. Several sentences were changed or eliminated or added in order to improve the text. All corrections are indicated in red in the text.
Furthermore, new recent bibliographic references have been added as follows:
Leogrande, R.; Lopedota, O.; Vitti, C.; Ventrella, D.; Montemurro, F. Saline water and municipal solid waste compost application on tomato crop: effects on plant and soil. J Plant Nutr Soil Sc. 2016, 39(4), 491-501 doi: 10.1080/01904167.2015.1084325.
FAOSTATS 2022. Available online: https://www.fao.org/faostat/en/#search/almond (accessed on 23 March, 2024).
Rouphael, Y., and Colla, G. Biostimulants in agriculture. Front. Plant Sci. 2020, 11, 40. doi: 10.3389/fpls.2020.00040.
Weisser, M.; Mattner, S.W.; Southam-Rogers, L.; Hepworth, G.; Arioli, T. Effect of a fortified biostimulant extract on tomato plant productivity, physiology, and growing media properties. Plants 2024, 13, 4. https://doi.org/ 10.3390/plants13010004.
Hellequin, E.; Monard, C.; Chorin, M.; Daburon, V.; Klarzynski, O.; Binet, F. Responses of active soil microorganisms facing to a soil biostimulant input compared to plant legacy effects. Sci. Rep. 2020, 10, 1–15.
Della Lucia, M.C.; Bertoldo, G.; Broccanello, C.; Maretto, L.; Ravi, S.; Marinello, F.; et al. Novel effects of leonardite-based applications on sugar beet. Front. Plant Sci. 2021, 12:646025. doi: 10.3389/fpls.2021.646025.
Distefano M.; Mauro, RP.; Page D.; Giuffrida, F.; Bertin, N.; Leonardi, C. Aroma volatiles in tomato fruits: the role of genetic, preharvest and postharvest factors. Agronomy. 2022. 12, 376. doi:10.3390/agronomy12020376.
Riga, P.; Anza, M.; Garbisu, C. Tomato quality is more dependent on temperature than on photosynthetically active radiation. J. Sci. Food Agric. 2008, 88, 158–166. doi: 10.1002/jsfa.3065
4) The reviewer wrote: outline 2.4 need to change into (Furit harvesting and qualitative analysis.
Our answer: We changed the outline 2.4 in Fruit harvesting and qualitative analysis”.
5) The reviewer wrote: Please must revise all tables, not in proper formate.
Our answer: We have reviewed all the tables and formatted them according to the style of the journal Plants.
6) The reviewer wrote: I suggest change Table 3 into Figure. Readers will easily understand.
Our answer: As regards the climate tables, we kindly ask you to leave the tables 3, as we would have great difficulty replacing them with graphs
7) The reviewer wrote: Experiment 1 and experiment 2 is not a proper and scientific wording. I suggest to change it.
Our answer: We changed the word Experiment to Trial
8) The reviewer wrote: Also discussion need improvement, I suggest to add sub-headings, and need to add more explaination for the support of your results.
Our answer: As you requested the discussion has been improved
9) The reviewer wrote: I suggest to add the results of correlation analysis.
Our answer: Correlation analysis has been added:
A linear positive relationship was found between the SPAD index and marketable yield (R = 0.726)
10) The reviewer wrote: Conclusion is well drawn but may i know, how your work is important for a local farmer, and for an industralist?
Our answer: In the conclusions we have added the following sentence:
“Ultimately, the results of this study in terms of yield mentioned above can be useful to assist growers in the fertilization practice of tomato crops and ensure product quality for industrialists”. We send you sincere greeting
Grazia Disciglio, Annalisa Tarantino and Laura Frabboni
Reviewer 4 Report
Comments and Suggestions for Authors
The work exhibits remarkable clarity and structure, and its results are both novel and interesting regarding the increase in crop production through the application of new biostimulants. However, there are aspects that could be improved, such as comparing the application of all fertilizers in both types of soils to evaluate effectiveness in different edaphological contexts. Additionally, it would be enriching to include SPAD values in the second experiment with fertilizer application in the soil, as it could provide a greater understanding of its relationship with production.
Corrections and suggestions to improve the manuscript:
- Correct the numerical format in line 106 from 2.1‰ to 2.1%.
- Ensure that the units in Table 2 are consistent for all fertigation applications.
- Suggest conducting SPAD measurements from the beginning of the experiment.
- Correct Figure 4, as the results of Trial B have been placed instead of Trial A. Additionally, add significance to the results in the figure.
- It is suggested to explore possible reasons for the differences in results between the two experiments (Trial A Field 1 (2021) and Trial B Field 1 (2023). A decrease in yield was observed in the second experiment, in contrast to the first year. Similarly, a reduction in fruit weight was recorded. However, certain physico-chemical characteristics of the tomatoes showed a slight increase in the second experiment compared to the first. For example, Brix, total phenols, or lycopene concentration.
- It would be valuable to include SPAD measurements in the second experiment to validate its correlation with yield, as was done in the first experiment.
- It is recommended to justify the statement that tomato quality is more influenced by temperature than by photosynthetically active radiation.
- It would be interesting to indicate which biostimulants are considered optimal for regular use in tomato cultivation, taking into account the increased production costs.
- It is recommended to identify the soil characteristics that could limit the use of biostimulants and to discuss which soils would be most suitable for their application.
In summary, addressing these considerations would enhance the study and make it more suitable for publication, providing a more comprehensive and precise insight into the effects of biostimulants on tomato production.
Author Response
Dear Reviewer,Thanking you for your appreciation of the manuscript and for the comments to which we respond and insert new sentences as follows:
- The reviewer wrote: Correct the numerical format in line 106 from 2.1‰ to 2.1%.
Our answer: We have corrected the unit of measurement
- The reviewer wrote: Ensure that the units in Table 2 are consistent for all fertigation applications.
Our answer: The units in Table 2 (became tab.3) have been standardized for all fertigation applications.
- The reviewer wrote: Suggest conducting SPAD measurements from the beginning of the experiment.
Our answer: The suggestion is accepted.
- The reviewer wrote: Correct Figure 4, as the results of Trial B have been placed instead of Trial A. Additionally, add significance to the results in the figure.
Our answer: We have inserted the correct figure 4 with significance regarding the Trial A.
- The reviewer wrote: It is suggested to explore possible reasons for the differences in results between the two experiments (Trial A Field 1 (2021) and Trial B Field 1 (2023). A decrease in yield was observed in the second experiment, in contrast to the first year. Similarly, a reduction in fruit weight was recorded. However, certain physico-chemical characteristics of the tomatoes showed a slight increase in the second experiment compared to the first. For example, Brix, total phenols, or lycopene concentration.
Our answer: To explain the differences between years, we have included the following sentence:“The lower yield obtained in 2023 compared to that of 2021 may have been caused by the excessive maximum temperatures recorded in July of the same year (on average 37.9 °C, but with values up to 44 °C on some days), creating stress on the crop during the fruit enlargement and ripening phases, resulting in a reduction in the weight of the fruits, albeit with a slight improvement in their qualitative characteristics”.
- The reviewer wrote: It would be valuable to include SPAD measurements in the second experiment to validate its correlation with yield, as was done in the first experiment.
Our answer: We are sorry, but the SPAD measurements were carried out only in the first experiment as it was physically not possible to carry them out in the second experiment.
- The reviewer wrote: It is recommended to justify the statement that tomato quality is more influenced by temperature than by photosynthetically active radiation.
Our answer: The above statement was reported in the article by Riga et al., as a result of a greenhouse experiment to study the effect of cumulative air temperature and photosynthetically active radiation (PAR) on tomato quality.
We have added the following sentence and reference to the text:
Finally, as reported by Riga et al. [73], tomato quality is more dependent on temperature than on photosynthetically active radiation.
Riga, P.; Anza, M.; Garbisu, C. Tomato quality is more dependent on temperature than on photosynthetically active radiation. J. Sci. Food Agric. 2008, 88, 158–166. doi: 10.1002/jsfa.3065
- The reviewer wrote: It would be interesting to indicate which biostimulants are considered optimal for regular use in tomato cultivation, taking into account the increased production costs
Our answer: In the conclutions, we have included the following sentence:
“Furthermore, taking into account the higher production costs, Foutroot® organo-mineral fertilizers or the combination of [Radicon®+Amifort-Plus®] applied to soil by fertigation are considered optimal for regular use in conventional agricultural practices of processing tomato crop”.
- The reviewer wrote: It is recommended to identify the soil characteristics that could limit the use of biostimulants and to discuss which soils would be most suitable for their application.
Our answer: In the conclutions, we have included the following sentence:
Additionally, they are more effective in soil conditions of poor fertility and low organic matter content.
- The reviewer wrote: In summary, addressing these considerations would enhance the study and make it more suitable for publication, providing a more comprehensive and precise insight into the effects of biostimulants on tomato production.
Our answer: We accepted all the considerations suggested by the reviewer and we sincerely thank them.
We send you sincere greeting
Grazia Disciglio, Annalisa Tarantino and Laura Frabboni
Round 2
Reviewer 1 Report
Comments and Suggestions for Authors
This study evaluated the effect of fertilization treatment on yield and quality of tomato based on two experimental studies. Although the author has made some revisions and improvements, there are still some key problems in the manuscript.
The two trials were not closely related and were conducted at different sites. As a result, the research results and conclusions are simply superimposed by two experiments, and there is no substantial hypothesis and design to solve the unified problem through different experiments.
Line17, What does radicon mean?
Line17, What does SPAD refer to? The first mention needs the full name.
Line45, “tomato” is ok, not need express as “tomato crops”.
Line81, “However, little research has been conducted under conventional nutritional conditions”. In fact, there are many related studies, and this is not the reason to carry out this study.
Line118, The special symbols in Table 2 need to indicate the specific meaning.
Line141, All are declarative content and do not require table presentation.
Line183, Almost no climate conditions were involved in the analysis of the results of this study, so it is not necessary to present relevant parameters in such detail in the text.
Line228, “3.1. Trial A: Fields 1 and 2” and “3.1.1. SPAD Index Analysis” is not suitable as a subheading for the results section; it requires a specific result statement as a subheading.
On the whole, although this study has carried out some experimental treatment, the analysis method is simple, and the scientific value of the results is limited.
Comments on the Quality of English LanguageThe English language needs further editing.
Author Response
Dear Reviewer,
Thanking you for your comments and we apologize for some of our inaccuracies we respond as follows:
The reviewer wrote: The two trials were not closely related and were conducted at different sites. As a result, the research results and conclusions are simply superimposed by two experiments, and there is no substantial hypothesis and design to solve the unified problem through different experiments. Our answer: As regards the correct considerations of the reviewer, it was fundamental for us to specify the particularities of the research in the conclusions.Therefore, by virtue of this, the conclusions have been rewritten with the following additions:
“It should be considered that, unless part of the research concerns the comparison between the treatment with Radicon® conducted in the same field, it is not possible to draw unifying conclusions between the two trials conducted in different sites. However, for each of them the results provided very interesting indications. In general, biostimulant treatments through application of the Radicon® product by foliar or by fertigation increased the yield compared to the untreated control, while slight effects were found on some qualitative attributes of tomato fruits. Furthermore, through the foliar application of the Radicon® product, the best production results were obtained in low fertility soil conditions and there was a significant increase in the total chlorophyll pigment content compared to the control. The soil application of Futuroot® or the combination [Radicon® + Amifort-Plus®) products containing not only biostimulant compounds but also auxins, cytokinins and microelements (Z, Mg, Mn), also increased yield compared to the untreated control, while slight effects were found on some qualitative attributes of tomato fruits”.
The reviewer wrote: Line17, What does radicon mean?
Our answer: To explain the meaning of Radicon ® in line 17 we have inserted the names of the organo-mineral products used in the previous sentence (line 13), as follows:
“Two independent trials were conducted across two seasons (2021 and 2023) to evaluate the effects of foliar or soil applications of various commercial organo-mineral fertilizers (Futuroot®, Radicon® Amifort®) with biostimulant action on the
The reviewer wrote: Line17, What does SPAD refer to? The first mention needs the full name. Our answer In line 17 SPAD is referred to as follows: “the leaf green color intensity”
The reviewer wrote: Line45, “tomato” is ok, not need express as “tomato crops”.
Our answer: We have eliminated the word "crops"
The reviewer wrote: Line81, “However, little research has been conducted under conventional nutritional conditions”. In fact, there are many related studies, and this is not the reason to carry out this study.
Our answer: the concept has been changed by inserting the following sentence:……”several also conducted under conventional nutritional conditions [17, 38, 39], but comparative research on fields with different fertility is lacking”.
The aim of this research was……
The reviewer wrote: Line118, The special symbols in Table 2 need to indicate the specific meaning. Our answer We have inserted the specific meaning of the symbols at the back of table 2 as follows:
“The "ø" denotes diameter; W/V means Weight/Volume”
The reviewer wrote: Line141, All are declarative content and do not require table presentation.
Our answer We eliminated the table presentation data in the text.
The reviewer wrote: Line183, Almost no climate conditions were involved in the analysis of the results of this study, so it is not necessary to present relevant parameters in such detail in the text. Our answer: Of the climatic parameters in Table 4 not discussed, we have eliminated the data relating to humidity, while the data relating to wind have been left as the high windiness is a peculiar characteristic of the site. For this reason the following sentences have been added to the text and the reduced table has been inserted:
“and further by the strong speed wind”.
“The average daily wind speed varied between 3.3 and 4.3 m sec-1“
|
Month |
Tmax |
Tmin |
Ws |
P |
EV |
Kc |
|
|
(°C) |
(°C) |
(m sec-1) |
(mm) |
(mm) |
(EV) |
|
|
|
|
2021 |
|
|
|
|
April |
19.9 |
4.7 |
4.3 |
40.4 |
161,3 |
0.35 |
|
May |
26.5 |
10.8 |
3.5 |
26.0 |
203.2 |
0.50 |
|
June |
33.2 |
15.9 |
3.3 |
8.6 |
230.2 |
0.95 |
|
July |
35.4 |
19.3 |
3.7 |
100.8 |
247.8 |
1.03 |
|
August |
34.9 |
19.4 |
3.8 |
29.2 |
238.9 |
0.80 |
|
September |
29.5 |
15.4 |
3.5 |
19.4 |
192.4 |
---- |
|
Mean |
29.9 |
14.3 |
3.7 |
|
|
0.73 |
|
Total |
|
|
|
224.4 |
1273.8 |
|
|
|
|
|
2023 |
|
|
|
|
April |
18.0 |
8.1 |
3.5 |
66.2 |
162.8 |
---- |
|
May |
24.0 |
11.9 |
3.5 |
73.6 |
200.2 |
0.40 |
|
June |
31.1 |
16.0 |
3.8 |
84.0 |
225.9 |
0.78 |
|
July |
37.9 |
19.9 |
3.5 |
3.2 |
255.5 |
1.06 |
|
August |
34.6 |
18.4 |
4.2 |
28.4 |
218.0 |
0.76 |
|
September |
30.6 |
17.4 |
4.0 |
17.6 |
204.0 |
0.60 |
|
Mean |
29.4 |
15.3 |
3.8 |
|
|
0.72 |
|
Total |
|
|
|
273.0 |
1266.4 |
|
The reviewer wrote: Line228, “3.1. Trial A: Fields 1 and 2” and “3.1.1. SPAD Index Analysis” is not suitable as a subheading for the results section; it requires a specific result statement as a subheading.
Our answer: As suggested, we eliminated “SPAD Index Analysis” as subheading for the results section
.
The reviewer wrote: On the whole, although this study has carried out some experimental treatment, the analysis method is simple, and the scientific value of the results is limited.
Our answer: As already indicated previously, in order to better clarify the overall results of this study and the use of the experimental treatments, the following is additionally specified in the conclusions:
“It should be considered that, unless part of the research concerns the comparison between the treatment with Radicon® conducted in the same field, it is not possible to draw unifying conclusions between the two trials conducted in different sites. However, for each of them the results provided very interesting indications. In general, biostimulant treatments through application of the Radicon® product by foliar or by fertigation increased the yield compared to the untreated control, while slight effects were found on some qualitative attributes of tomato fruits. Furthermore, through the foliar application of the Radicon® product, the best production results were obtained in low fertility soil conditions and there was a significant increase in the total chlorophyll pigment content compared to the control. The soil application of Futuroot® or the combination [Radicon® + Amifort-Plus®) products containing not only biostimulant compounds but also auxins, cytokinins and microelements (Z, Mg, Mn), also increased yield compared to the untreated control, while slight effects were found on some qualitative attributes of tomato fruits”.
Sincere greeting,
Grazia Disciglio, Annalisa Tarantino and Laura Frabboni

Reviewer 3 Report
Comments and Suggestions for Authors
Author has addressed all comments and thus the revised version has been improved and accepted for publication.
Author Response
Cover letter for the Reviewer 3
Dear Reviewer,
We thank you for accepting the revised and improved version for publication
Sincere greeting,
Grazia Disciglio, Annalisa Tarantino and Laura Frabboni